# Comparative and functional genomic analysis of foreign DNA defense mechanisms in *Enterococcus faecium*

Alexandra L. Krause,[1] Louise M. Judd,[1,2] Ryan Wick,[1,2] Timothy P. Stinear,[1,2] Andrew H. Buultjens,[1] Ian R. Monk[1]

**ABSTRACT** *Enterococcus faecium* is notoriously difficult to study genetically due to the poor understanding of barriers preventing foreign DNA uptake, such as the proteins that modify type I restriction modification (RM) system activity. Here, we compared *E. faecium* repertoires of the HsdS specificity subunit (dictating the DNA motif that is adenine methylated) from type I RM systems among 805 globally reported *E. faecium* isolates. We showed there were eight distinct HsdS types, with four dominant variants that were also significantly enriched in the hospital-associated clade A1 *E. faecium* lineage. Adenine methylome analysis of a subset of eight representative *E. faecium* strains revealed that only two exhibited functional type I RM systems, with the activity corroborated by the construction of type I RM deletion mutants. To investigate this surprising finding, we assessed the contribution of the anti-restriction protein ArdA that specifically inhibits type I RM function. The *E. faecium* ST796 clinical isolate AUS0233 has one intact type I RM system, no adenine methylation, and two distinct *ardA* paralogs. When heterologously expressed in *Staphylococcus aureus* JE2, both *E. faecium ardA* variants were functional, each inhibiting the function of the two type I RM systems in *S. aureus*. However, the deletion of one or both versions of *ardA* in *E. faecium* AUS0233 did not change the transformation efficiency with exogenous DNA, suggesting ArdA in *E. faecium* AUS0233 is not controlling type I RM. This study highlights the complexity of DNA defense mechanisms in *E. faecium* and suggests that unidentified factors control the acquisition of foreign DNA.

**IMPORTANCE** *Enterococcus faecium* has mechanisms of DNA methylation and targeted DNA degradation (called restriction modification [RM]) that hinder foreign DNA uptake, thus influencing the acquisition of important phenotypes such as antibiotic resistance. Restriction barriers also frustrate efforts for laboratory genetic manipulation used to study this pathogen. From PacBio analysis of *E. faecium* strains, it was observed that the majority of *E. faecium* do not adenine methylate DNA despite genome analysis indicating they have intact type I RM methylation systems. One explanation for this observation is that *E. faecium* produces anti-restriction factors such as ArdA, which can inhibit type I RM systems. However, the deletion of both *ardA* alleles did not improve the efficiency of DNA uptake. These findings build our foundational knowledge of how *E. faecium* controls foreign DNA and show there is additional complexity surrounding these systems to be discovered.

**KEYWORDS** *Enterococcus faecium*, genomics, DNA defense mechanisms, restriction modification system, *hsdS*

Enterococci are a dominant member of the human gut microbiome (1, 2). However, two species in the genus, *Enterococcus faecium* and *Enterococcus faecalis*, have become notable pathogens causing nosocomial infections (3). Once the intestinal epithelial layer has been breached, they can subsequently enter the bloodstream and

**Peer Reviewer** Willem Van Schaik, University of Birmingham, Birmingham, West Midlands, United Kingdom

Address correspondence to Ian R. Monk, imonk@unimelb.edu.au, or Timothy P. Stinear, tstinear@unimelb.edu.au.

The authors declare no conflict of interest.

See the funding table on p. 17.

cause persistent infections due to their resistance to a wide range of antibiotics (3). Vancomycin-resistant *Enterococcus faecium* (VREfm) is particularly difficult to treat (4, 5), with high rates of resistance to ampicillin, among other antibiotic classes (6).

Comparative population genomics has shown that *E. faecium* can be divided into two major clades: clade A, being the clinical, hospital endemic strains, and clade B (now known as *Enterococcus lactis* [7]) containing the commensal, generally more antibiotic-sensitive strains (8). Clade A further divides into A1 and A2, with clade A2 associated with *E. faecium* from animal reservoirs, while clade A1 consists of more hospital-adapted *E. faecium* clones (9). Recent genomic comparisons have strengthened the presence of two clade A subgroups (10). A potential explanation for the distinct clade structure is the presence of restriction modification (RM) systems controlling exchange of mobile genetic elements (MGE) between strains (11, 12). The ability for strains to pass MGE between populations is an important mechanism for the acquisition of antibiotic resistance, virulence, and metabolic genes, which could be vital for survival in certain niches (13). More often, the A1 hospital-adapted strains exhibit increased plasmid content compared to the A2 and B clades, with this plasmidome shown to improve their fitness in the hospital environment (12).

RM systems provide a barrier to the entry of foreign DNA and infection by bacteriophage (11). There are four main classes of restriction modification systems; however, only type I, type II, and type IV have been described in *E. faecium* and *E. faecalis* (14). Type I RM systems are comprised of three protein subunits, termed host specificity of DNA (HsdS), specificity (S), modification (HsdM—methylase) protein, and the restriction endonuclease (HsdR) (14). The HsdM and HsdS form a protein complex ($HsdM_2 HsdS_1$) that adenine methylates host DNA. The sequence methylated is dictated by the two-target recognition domains (TRD) of the HsdS, which recognize an asymmetric bipartite DNA sequence. The two HsdS TRD motifs comprise 3–4 base pairs, which contain the methylated adenine residue, separated by 4 to 9 non-specific base pairs. During DNA replication, the newly replicated strand is adenine methylated at the HsdS-dictated TRD motif in the presence of hemi-methylated template. Two HsdR subunits combine with the $HsdM_2HsdS_1$ complex in the absence of a specific adenine methylation pattern, transforming the system into a molecular motor bound to the DNA, introducing DNA breaks through collision with either a second bound RM complex, other proteins, or DNA secondary structure. Additionally, CRISPR-Cas (present in ~75% *E. faecalis* isolates, but only in ~5% of *E. faecium* clade A strains) can limit horizontal gene transfer (HGT) of MGE (8, 15). Other elements, such as cyclic-oligonucleotide-based anti-phage signalling systems, abortive infection, or ArdA (mimics the shape and structure of a short DNA sequence to inhibit type I RM activity), could play a role in genome defense through discrimination of foreign from host DNA (16–18).

RM systems and other genome defense mechanisms pose significant challenges to genetic manipulation in the laboratory. These barriers complicate efforts to study clinical *E. faecium* strains at a molecular level, such as identifying novel drug targets or uncovering the mechanisms underlying hospital adaptation. The application and confirmation of forward genetic screens in clinical isolates is reliant on bypassing this obstacle. In this study, we combined *in silico* and functional methylome analysis to define the DNA defense systems present in a global selection of clinical *E. faecium* isolates and determine the functionality of the type I RM systems present. We examined the global presence and function of HsdS alleles in select isolates. Furthermore, we defined the impact of the anti-restriction protein, ArdA, on type I RM system function by assessing transformation efficiency in *E. faecium* AUS0233 and *Staphylococcus aureus* JE2.

## RESULTS

### A globally diverse panel of 20 clade A1 *E. faecium* isolates

To gain insight into the recalcitrant nature of clinical *E. faecium* isolates to genetic manipulation, a diverse test panel of 20 *E. faecium* clade A1 strains (sensitive to chloramphenicol) was selected to perform transformation experiments. These strains

were isolated from patients between 2006 and 2018, across four clinically relevant multi-locus sequence types (MLSTs) (ST80, ST203, ST796, and ST1421) with either VanA or VanB vancomycin resistance loci. (Fig. 1). Finished genome sequences for these 20 isolates were established and compared with a selection of 785 publicly available draft genomes spanning clades B, A2, and A1 (10) to contextualize the test panel and extrapolate analyses in a global context (Table S1; Fig. 1).

## Transformation efficiency of the 20-strain panel

Experiments were conducted with the 20 *E. faecium* strains selected above to identify the optimal glycine concentration to obtain maximal transformation efficiency under the test conditions. Addition of glycine to growth media has previously been shown to cause cell wall weakening, allowing enhanced DNA uptake (19). Electrocompetent cells were generated and transformed with the 5.5 kb plasmid pIMC8(P*gap*-YFP). Transformants were observed for all the strains tested, with half the strains yielding between $5 \times 10^2$ and $5 \times 10^3$ CFU/µg DNA. Two strains were poorly transformable (2394 and 2397) and did not yield any transformants in two out of the three replicate experiments. Strains 2395 and 2396 contain a total of 150 SNP differences and exhibited very similar transformation efficiencies. Consistent with the literature, this panel of *E. faecium* strains

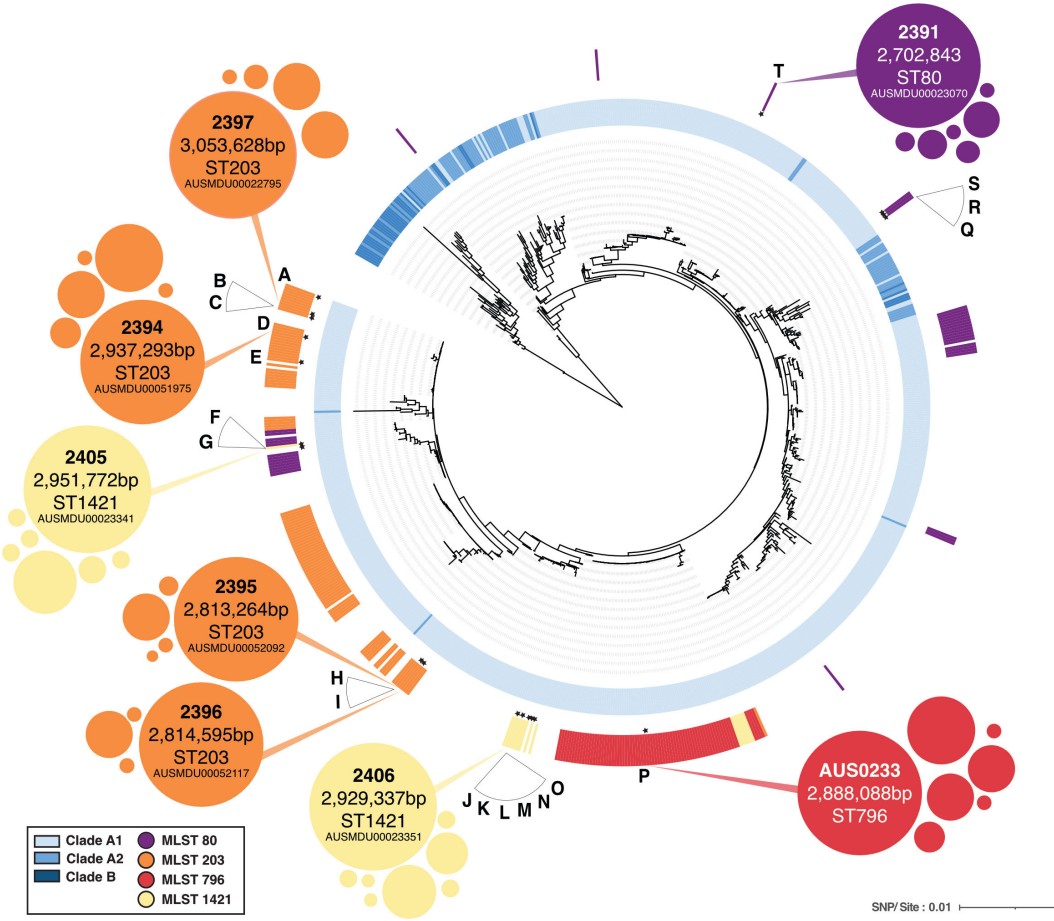

**FIG 1** Global core-single nucleotide polymorphism (SNP) based phylogeny of 805 *E. faecium* strains. The 805 strains are representative of 121 different MLSTs across clade B (dark blue), clade A2 (blue), and clade A1 (light blue). Sequence types ST80 (light orange), ST203 (orange), ST796 (red), and ST1421 (yellow) are colored. The 20 test panel strains are highlighted on the tree by stars, and the letter code (A to T) is linked to the test panel strain ID in Table S2. The phylogenetic positions of eight PacBio-sequenced isolates are overlaid onto the phylogenetic tree. The large bubble represents the chromosome, and the smaller surrounding bubbles illustrate the plasmid content of each strain, with the size being a semi-quantitative depiction of plasmid size. This tree was recombination filtered using *Clonal Frame ML,* and nodes with less than 70% bootstrap support were collapsed. Branch distance is shown as SNP per site.

was less transformable than *E. faecalis,* which ranged between $1 \times 10^5$ CFU/µg DNA V583 (Fig. 2A) and $1 \times 10^6$ CFU/µg DNA for JH2-2 (19).

## Genomic analysis of DNA defense systems

To identify the barriers to DNA uptake present in the *E. faecium* panel, we ran *in silico* analysis of the genomes to predict the presence of RM systems using "DefenseFinder," a tool designed for identifying anti-phage defense mechanisms (20, 21) (Fig. 2B). We observed that all strains except 2391 contained at least one type I RM system. Strain 2391 contained a novel type IIG RM system (Fig. 2B). The co-presence of an adenine methylating type IIG and type I system might be incompatible due to the potential for overlap in the adenine sites methylated. From the other 19 strains, four different HsdS alleles were identified (HsdS_1 to HsdS_4), with HsdS_4 solely plasmid-associated, located on a >200 kb plasmid in strains 2395, 2396, 2405, and 2406 (closed genomes assembled—see below). HsdS_4 was detected in 5/20 strains with draft genomes, and the plasmid association was confirmed through the absence of SNPs when the HsdS_4 allele-containing contig was mapped to the fully assembled 2405 HsdS_4-containing plasmid. The other HsdS alleles were located on the chromosome, often flanked by transposable elements (Fig. 2C). The HsdS_1 allele was found in three out of four ST80 isolates and two out of seven ST203 but was truncated through the insertion of a transposase in the ST80 isolates examined. The four HsdS alleles identified here were previously shown to be enriched in clade A1 isolates (12), with HsdS_1 the first *E. faecium* HsdS allele to be characterized (11).

## Evaluation of methylation activity of RM systems through PacBio sequencing

To assess the functionality of the RM systems, eight isolates were selected for PacBio sequencing (Fig. 1) (22, 23). The strains were chosen either due to their low transformation efficiency, suggestive of active RM, or the combination of type I RM systems present in the strain. PacBio sequencing can identify consensus adenine methylation (m6A), which permits attribution of the HsdS allele with its cognate recognition sequence (24). Genomes for *E. faecium* AUS0233, 2391, 2394, 2395, 2396, 2397, 2405, and 2406 were assembled using PacBio long reads and polished with Illumina reads (25). These eight strains are from ST80, ST203, ST796, and ST1421 and are genetically diverse among clade A1 isolates, as shown in the phylogenetic tree of global *E. faecium* isolates (Fig. 1). Surprisingly, m6A methylation could only be detected in three of the eight strains (Fig. 2C). Strain 2405 had a consensus m6A profile identified 5′ CYYANNNNNNTGAY 3′ with 90%–91% of sites methylated. Searching the REBASE database of known recognition motifs for HsdS proteins with HsdS_4 yielded an HsdS allele with the same consensus motif (26). A second consensus m6A profile of 5′ T**A**CNNNNNNNGTA 3′ was also identified, which was methylated at 81% of 810 sites, but only one DNA strand of the motif was methylated. This recognition motif correlates with the HsdS_2 allele, although not present on REBASE, and could be attributed to elimination. We suspect that the unique palindromic nature of the recognition motif prevented differentiation of m6A strand methylation. A type IIG RM system was found in strain 2391 with an m6A signature of 5′ GG**A**GG 3′ with methylation present on 2094/2996 sites (27). A type IIG system has not been identified in *E. faecium* before, but it does not appear to greatly impact transformation efficiency (Fig. 2A). The plasmid [pIMC8(P*gap*-YFP)] used to test transformation efficiency contains eight potential methylation sites. However, it is also possible that the restriction function of the type IIG system might be compromised. Additionally, m6A was detected in strain 2394 with a recognition motif of 5′ G**A**CNNNNNNNGTA 3′ (HsdS_3); however, only 23%–26% of the 705 sites in the genome were methylated.

## Comparative analysis of type I RM systems in *E. faecium* strains

Alignment of the HsdM and HsdR proteins obtained from the four type I RM systems with confirmed methylation activity (*E. faecium* 1,231,502—HsdS_1 (11), 2405 HsdS_2/

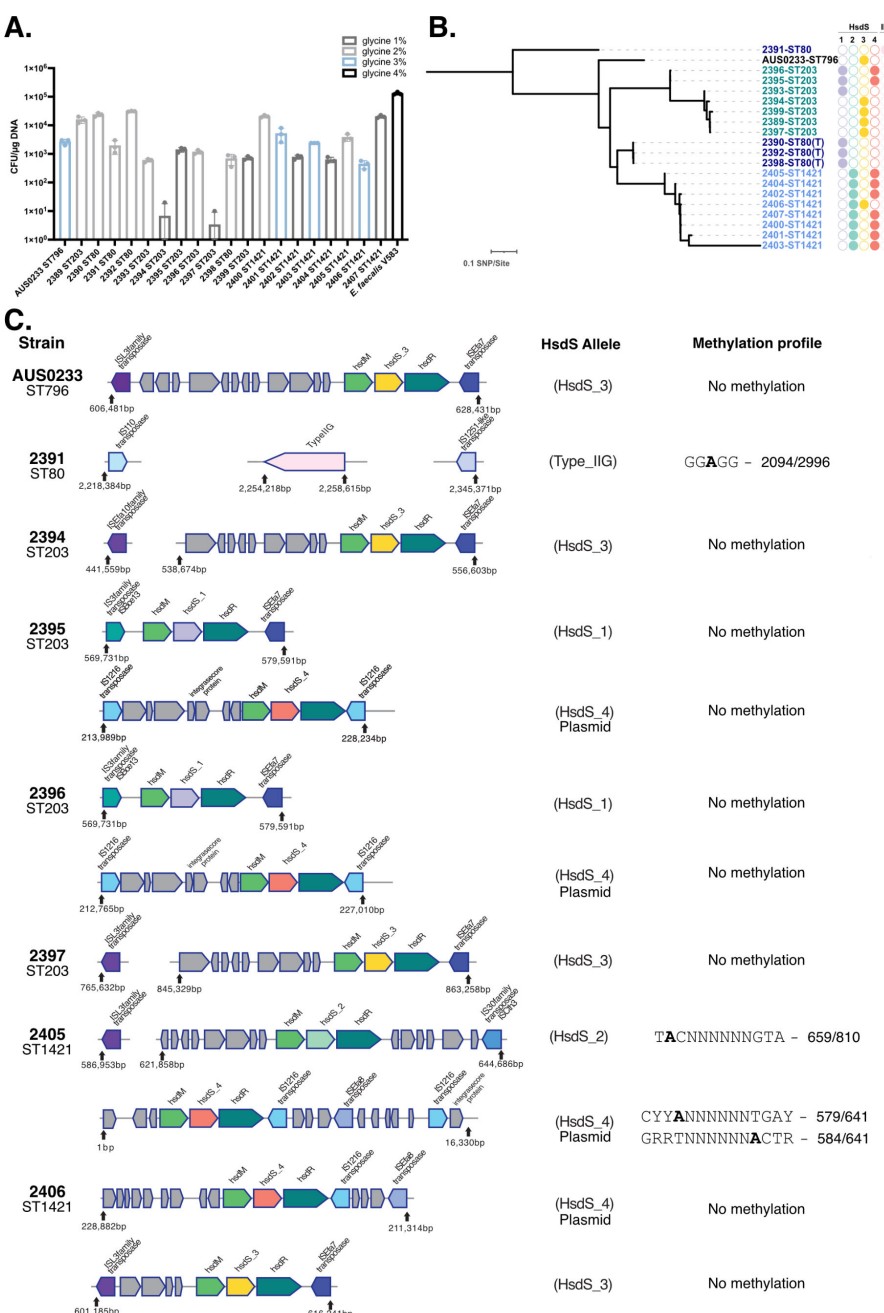

**FIG 2** Examination of type RM systems found in the 20-strain *E. faecium* panel. (A) A panel of 20 clinical *E. faecium* strains was transformed with plasmid pIMC8(P*gap*-YFP) (1 µg), with the optimal glycine percentage used shown. *E. faecalis* strain V583 is shown as a comparator. The data is from three independent batches of competent cells transformed, with the mean and standard deviation (SD) shown. (B) Rooted core-SNP-based phylogenetic tree of VRE strains with the presence of HsdS alleles shown, purple (HsdS_1), green (HsdS_2), yellow (HsdS_3), and red (HsdS_4), (T) indicates a truncated HsdS protein (Table S4). The type IIG system is shown in pink. (C) Genomic organization of RM systems and methylation profiles of the eight PacBio sequenced strains. The methylated adenine residue in the methylation profile is highlighted in bold, with the fraction proceeding representing the adenine methylated versus the total number of sites across the genome. Genes are not drawn to scale.

HsdS_4, and 2394 HsdS_3), identified that minor differences in the amino acid sequence, in comparison to substantial variation observed in HsdS (Fig. S1). These differences were conserved across the panel of 20 isolates containing the same HsdS alleles, except for

strains 2395 and 2396 (HsdS_1), which contained an HsdR$^{P1026A}$ change. One caveat is that due to the presence of repeating DNA sequence features, contigs often break at the very 5′ end of *hsdM* and/or the very 3′ end of *hsdR* in Illumina-only assemblies, precluding complete analysis of the region. *E. faecium* strains AUS0233 and 2397 were both m6A$^-$ even though they contain an identical HsdS_3 chromosomal locus to strain 2394 (100% nucleotide match across the *hsdMSR* locus, including the flanking 9.8 kb upstream and 8.7 kb downstream) (Fig. S2), which corroborates a previous study (28). The pairwise alignment of the HsdS_4 type I RM system plasmid regions of 2405 (m6A$^+$) and 2406 (m6A$^-$) (Fig. S3A) also highlighted the absence of mutations between the identical type I RM systems in these strains (100% nucleotide match of 2405 *hsdMSR* locus and 2406 *hsdMSR* locus including the flanking 2.2 kb upstream and 1.6 kb downstream). Additionally, the plasmid alignments for 2405 (213 kb) and 2406 (229 kb) emphasize the similarity between plasmids that contain the type I RM system (HsdS_4) in both ST1421 strains (Fig. S3B). Gene presence-absence analysis of the 2405 and 2406 HsdS_4-containing plasmid identified five genes unique to 2405, including genes encoding three hypothetical proteins, a DNA topoisomerase III, and a transposase. In contrast, 23 unique genes were found in 2406, primarily related to metabolism and transposase function, rather than restriction-modification processes (Table S3B). A putative histidine kinase and a response regulator were present downstream of the type I RM system in the m6A$^-$ 2406, which was absent in the m6A$^+$ 2405. A precedent for two-component system (TCS) regulation of type I RM has recently been shown in *Helicobacter pylori* (29). The role of this plasmid-based TCS will require additional investigation. Further genomic analysis of strains 2394 and 2397, both ST203 with the same type I RM system (HsdS_3), indicated the presence of a phage in 2397 and an *ardA* allele (ArdA1) that was absent in 2394. ArdA is an anti-restriction protein that functions as a mimic of double-stranded DNA, binding to and blocking the activity of the type I RM system (18). The remaining genes identified were predominantly hypothetical, as well as integrase and transposase genes (Table S3A). This suggests that there could be additional mechanisms impacting the functionality or expression of type I RM systems in some *E. faecium* strains, such as ArdA systems.

## Restriction modification system functionality

To further understand the functionality of these restriction modification systems, we constructed deletions of the AUS0233 *hsdMSR* operon and the *hsdR* genes in strains 2394 and 2406. All three strains contain a sole *hsdS_3* gene, with the test plasmid containing two HsdS_3 recognition motifs. Deletion mutants in AUS0233 and 2406 showed no significant change in transformation efficiency compared to the wild type, which fits with the absence of m6A methylation in these strains. However, deletion of the *hsdR* gene in strain 2394 that has an active HsdS_3 m6A pattern increased the transformation efficiency 100-fold (Fig. 3).

## Global diversity of HsdS alleles in *E. faecium*

Recently, Arredondo-Alonso et al. (12) characterized the role that the "plasmidome" has played in the emergence of *E. faecium* as a nosocomial pathogen. From complete genome and plasmid sequencing of 62 isolates, they identified nine novel HsdS subunits, which contained HsdS_1 to HsdS_4 presented above and additional alleles HsdS_5, 6, 7, 10, and 13. Further analysis of the repository of HsdS alleles deposited (https://gitlab.com/sirarredondo/efaecium_population/-/blob/master/Files/rmsystem/s_subunits.faa?ref_type=heads) from the study identified a further six unique alleles, yielding HsdS_1 through to HsdS_14 (Fig. S4A). Each of the 14 HsdS amino acid sequences was used to produce a unique NCBI protein accession (Table S3). The amino acid MUSCLE alignment of the 14 HsdS highlighted the differences in the TRD for each of the alleles (Fig. S4B). We found that one allele from the collection (HsdS_10) was truncated (286aa) and lacked a C-terminal portion containing the majority of the second TRD and therefore was excluded from the analysis. From the analysis of the 805 geographically diverse isolates, five of the alleles (HsdS_9, 11, 12, 13, and 14)

were not found, leaving eight unique alleles. The HsdS_1 allele was found in 230 strains spanning 27 STs (Table S6 and S7), while the HsdS_2 allele was found in 87 strains from 20 different MLSTs. The HsdS_3 allele was found in 173 strains from 21 different STs, and HsdS_4 in 77 strains from 12 different STs. The remaining four alleles (HsdS_5, 6, 7, and 8) span 13 STs and were never associated with a second allele. No strains were observed to contain both HsdS_1 and HsdS_2 alleles, but strains containing HsdS_1 as well as either HsdS_3 or HsdS_4 were present. We found that the HsdS_1 allele and HsdS_3 allele were the most common HsdS types in the 805 genomes with the largest span of MLST types containing these HsdS subunits (Tables S6 and S7). For A1 isolates, only HsdS_5 ($n = 2$) and HsdS_8 ($n = 6$) were found in addition to alleles HsdS_1 to HsdS_4, highlighting the dominance of HsdS_1 to HsdS_4 in the A1 population, comprising 98.5% of all alleles identified. From the different MLSTs, only ST796 isolates showed 100% carriage ($n = 73$) of an HsdS allele (HsdS_3), from geographically diverse strains, with two strains also carrying the HsdS_4 allele. This emphasizes that specific HsdS alleles dominate in certain STs, but exchange of genetic material between different STs with the same or similar HsdS complement could occur. However, it is curious that a connection between ST and HsdS alleles exists, despite the observation that five out of the eight strains tested lack methylation.

We then examined the prevalence of the four main HsdS alleles in clade A1 strains from countries providing at least 20 isolates (including Australia, Denmark, Germany, Japan, Netherlands, Spain, UK, and USA) (Fig. 4A). It was observed that HsdS_1 was the most common allele globally, as it was dominant in five of the eight countries. Among strains from the UK, there is a split distribution (similar proportions were observed in the Australian, Japanese, and Spanish isolates) with 33.9% of strains having HsdS_1, 20.7% HsdS_2, 39.6% HsdS_3, and 13.2% HsdS_4 (Fig. 2B). The sole type IIG system identified in strain 2391 was found in only one additional Australian isolate; therefore, it appears to be rare in most populations (Table S5). The carriage of the eight HsdS alleles amongst the different *E. faecium* clades was investigated. A considerably lower rate of HsdS carriage was observed in clade B (10.5%) and clade A2 (12.8%), when compared to clade A1 where 73.7% of strains have at least one HsdS allele (Fig. 4B). A further

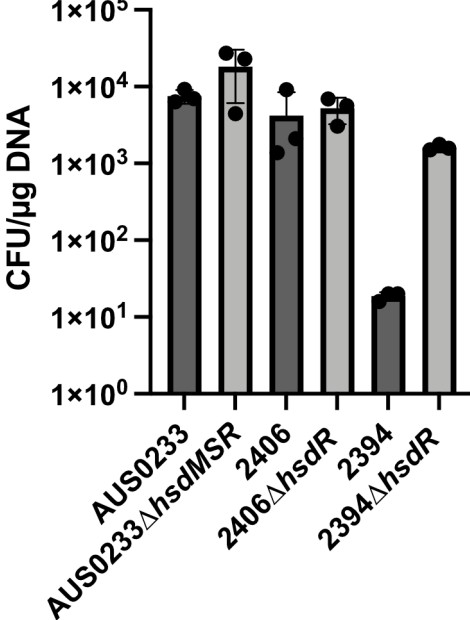

**FIG 3** Transformation efficiency of type I RM mutants. The wild-type and isogenic mutant were electroporated with 1 µg of pIMC8(P*gap*-YFP) isolated from *Escherichia coli* DC10B. The graph is representative of three independent transformations from different batches of competent cells, with the mean and SD shown.

reduction in the carriage of the dominant four alleles in clade B (7.9%) and clade A2 (4.7%) or 6.5% in non-A1 versus 73.1% in clade A1 was observed. A significant difference in HsdS allele proportions was observed between clade A1 and non-clade A1 strains (Chi-square test of independence, $P$-value < 0.001 and a Chi-square statistic of 266.7; degrees of freedom = 2) (30). The striking lack of HsdS alleles in non-clinical clades could be due to the possession of other DNA defense mechanisms, such as CRISPR-Cas, or the reduced selection pressure to acquire MGEs outside of the hospital environment (11, 12, 31). Conversely, the increased prevalence of HsdS in the clinical strains could help resist phage predation or, as previously suggested, shape the plasmidome of specific clones by allowing plasmid transfer between strains with compatible HsdS alleles (12). Dual HsdS allele carriage was also observed in the global collection, with 76/681 (11.1%) of the clade A1 strains containing two alleles (Table S6), but not identified in A2 or B strains. The HsdS pattern of the 20-test panel is slightly different from that observed in the 201 Australian clade A1 strains, with HsdS_1 being the dominant allele rather than HsdS_4 (45%). We have shown that HsdS_1 was the most common allele in the Netherlands, Denmark, Germany, and the USA, a predominance that might be due to the high proportion of clade A1 isolates sampled from these countries consisting of many clinically dominant STs such as 17, 78, 80, 203, 796, and 1421 (Table S7) (28, 32).

## Impact of ArdA expression on type I RM systems

We found that 76.7% of the 805 global *E. faecium* strains analyzed contain at least one of the four anti-restriction ArdA alleles identified in our test panel of 20 *E. faecium* strains (Fig. S5A and B). We observed the increased presence of ArdA alleles in strains from clade A1 (86.1%) compared to clade B (7.9%) and A2 (32.6%) (Fig. S5C). Additionally, we found that 382 strains have at least one ArdA and HsdS allele. There were 225 strains that lacked an HsdS allele but had at least one ArdA allele, and only 134 strains that had an HsdS allele without an ArdA allele. We found that 63 strains lacked both an ArdA and an HsdS allele.

We hypothesized that the presence of ArdA (33) in AUS0233 (which contains ArdA1 and ArdA2 alleles) could be contributing to the dysfunction (lack of m6A) of the type I RM system present. The two genes are distally located on the chromosome (0.64 and 2.76 Mbp), with ArdA1 and ArdA2 sharing 59% amino acid identity. ArdA1 is found on transposon Tn916, which contains *tetM*, and is located 15 kb upstream of the type I RM system (34). ArdA2 also appears to be on a transposable element similar, but not identical to Tn916 (35). Both *ardA* genes were sequentially deleted from AUS0233, and the transformation efficiency of the resulting mutants was compared to the wild type. There was no difference observed between the wild-type strain and the *ardA* mutants when transformed with pIMC8(P*gap*-YFP) (Fig. 5A), whereas if the ArdA proteins were impacting type I RM activity, we would have expected a reduction in transformation efficiency upon their deletion.

To further investigate the functionality of the anti-restriction proteins from AUS0233, we heterologously expressed the ArdA1 or ArdA2 proteins from the tetracycline-inducible expression vector pRAB11 (Fig. 5B and C) in *Staphylococcus aureus* JE2 (methicillin resistant, USA300 strain). Through heterologous expression, we compared the transformation efficiency of JE2 in the presence and absence of either ArdA allele. JE2 contains two well-defined type I RM systems as well as a type IV system (36). We used plasmid (pCN34*RBS, kanamycin resistance) extracted from either *Escherichia coli* IM08B, which expresses the m6A profile present in JE2 and is devoid of cytosine methylation (detected by the type IV system), or *E. coli* DC10B, which lacks cytosine methylation only. We found that when pCN34*RBS was extracted from IM08B (correctly adenine methylated), there was no difference in the number of transformants yielded (maximal transformation efficiency). However, the presence of either ArdA allele could inhibit the restriction activity of native type I systems, encoded by JE2. This was observed as a 2-$\log^{10}$ increase in transformation efficiency when the plasmid was isolated from DC10B, yielding a transformation efficiency equivalent to IM08B extracted plasmid. These data

**A.**

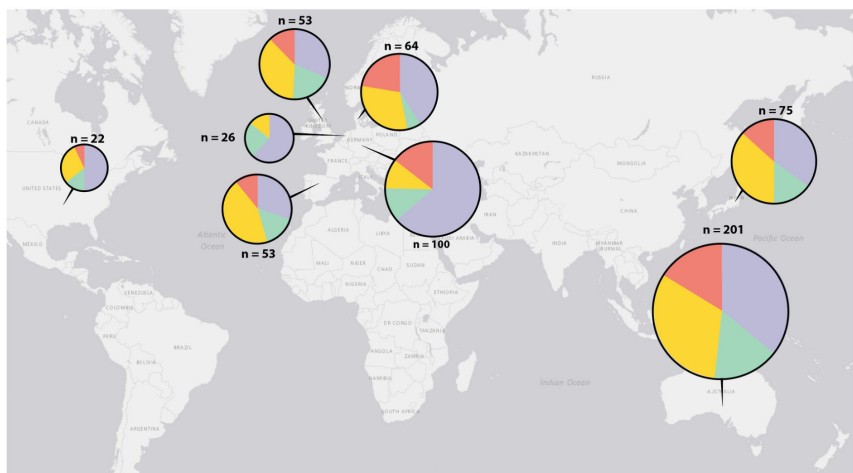

**B.**

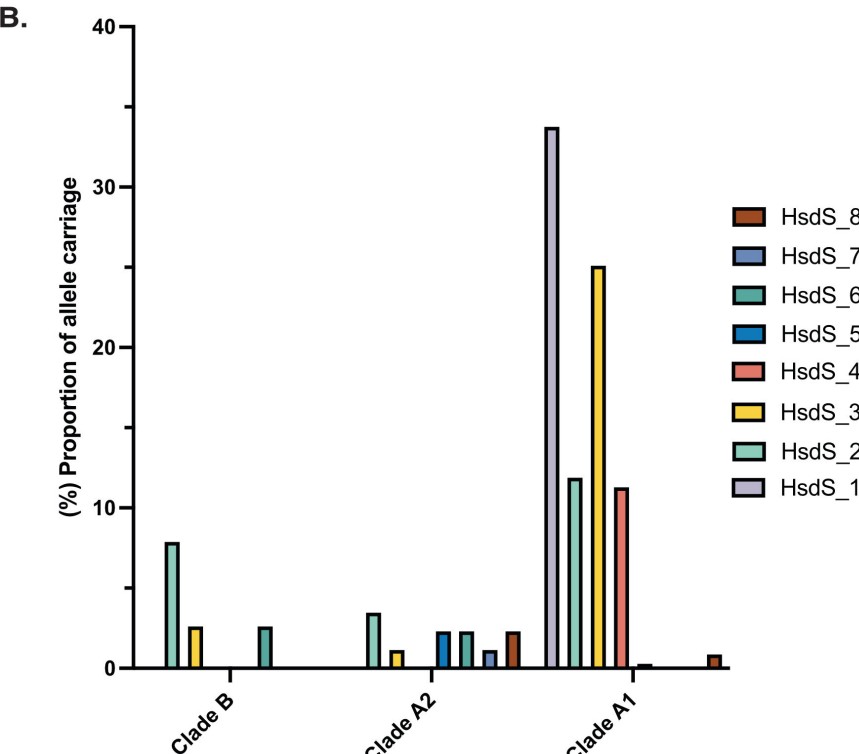

**FIG 4** (A) Global representation of the four main *hsdS* alleles: purple (HsdS_1), yellow (HsdS_2), green (HsdS_3), and red (HsdS_4). The pie charts represent the proportion of the four main types of HsdS for each country. Shown are the eight countries (out of 32) that had >20 strains from the clade A1 681 *E. faecium* genomes. Pie charts are size-scaled based on the number of strains present in the study. (B) Proportion of *hsdS* allele carriage in 805 global *E. faecium* strains. The strains are from clade B, clade A2, and clade A1, with HsdS subunits 1–8. These data are representative of 38 clade B strains, 86 clade A2 strains, and 681 clade A1 strains.

show that both ArdA alleles from AUS0233 are functional and can inhibit the restriction activity of type I RM systems *in vivo*, but raise the question of why the double *ardA* mutant did not exhibit reduced DNA uptake. Further experiments will be needed to measure the native levels of *ardA1/ardA2* expression in AUS0233 or the impact of ArdA overexpression in *E. faecium*.

## DISCUSSION

In this study, we investigated the genetic mechanisms causing the recalcitrant nature of clinical *E. faecium* to HGT, a feature that could be contributing to the increasing genetic divide between community and hospital strains of *E. faecium* (11). Through analysis of *hsdS* diversity in a global collection of 805 *E. faecium* genomes, we identified eight unique HsdS alleles, with 65.3% of genomes containing at least one. Uneven carriage between the clades was observed, with 13.2% of clade B and 12.6% of clade A2 containing an HsdS allele, compared to 73.7% of the A1 strains. Of the eight HsdS subunits, only the global top four were present in the 20-strain test panel (HsdS_1 to HsdS_4). Of the four, only HsdS_2 and HsdS_3 were sporadically identified in clades B and A2, suggesting limited overlap in clade carriage of HsdS alleles in *E. faecium* (Table S7). This contrasts with other Gram-positive bacteria, such as *S. aureus*, where combinations of HsdS alleles are specific to clonal complexes (37). However, a different picture emerges for *Staphylococcus epidermidis,* where type I RM systems are not conserved within sequence types, combined with more variation in HsdS (37). Additionally, it was found that 38% of *S. epidermidis* lacked a type I RM system, in line with our observations for *E. faecium*.

The possession of one or more of these four HsdS alleles could be beneficial for a strain, as it would enhance its ability to defend against bacteriophage infection (38). Additionally, allowing for controlled acquisition of MGEs from genetically similar isolates, which could assist with niche specialization (12, 39). Consequently, strains lacking a restriction modification system could have increased genetic diversity in populations as they are able to acquire any MGE regardless of methylation pattern. However, unless alternative DNA defense systems are present, such strains would be vulnerable to bacteriophage infection and invasion of foreign DNA that could damage genome integrity (38).

Even though the type I RM systems were present by *in silico* analysis, the functionality of these systems was unknown. When these 20 strains were tested for transformation efficiency, half of them exhibited an intermediate level of transformation ($5 \times 10^2$–$5 \times 10^3$ CFU/µg DNA), while two were extremely difficult to transform (Fig. 2A), which did not correlate with m6A status (2394; m6A$^+$, 2397; m6A$^-$). Through PacBio sequencing and consensus adenine methylation analysis, we showed that the majority of these type

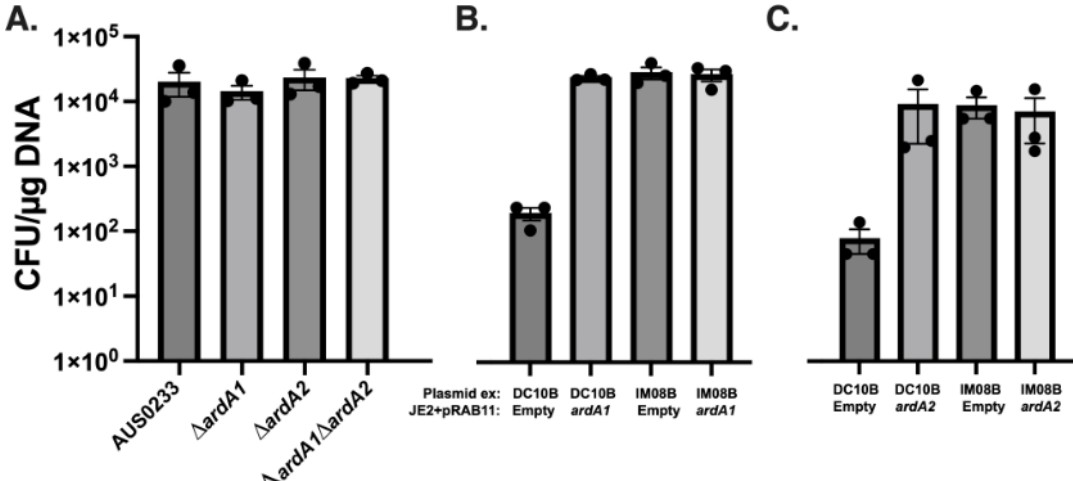

**FIG 5** Impact of ArdA on the transformation of *E. faecium* AUS0233 and *S. aureus* JE2. (A) The *ardA1* and/or the *ardA2* genes were deleted from AUS0233, and the transformation efficiency with plasmid pIMC8(P*gap*-YFP) was measured. (B) Role of *ardA1* or (C) *ardA2* on the inhibition of type I restriction in *S. aureus*. *S. aureus* JE2 was transformed with the tetracycline-inducible vector pRAB11 (empty) or pRAB11 containing (B) ArdA1 or (C) ArdA2. Electrocompetent cells of JE2+pRAB11 or JE2+pRAB11(*ardA1* or *ardA2*) were made in the presence of anhydrotetracycline (aTc, 100 ng/mL), and the transformation efficiency was determined with plasmid (pCN34*RBS) extracted from either DC10B (no adenine methylation) or *E. coli* IM08B (JE2 type I adenine methylation profile). These data are from three independent batches of competent cells transformed, with the mean and standard error of the mean shown.

I RM systems (7 out of 11 present) were not active, with only two (2394 and 2405) of the seven strains exhibiting type I m6A. To further characterize the type I RM activity in HsdS_3-containing strains (2394 [m6a$^+$] and AUS0233/2406 [m6a$^-$]), we deleted the *hsdR* genes and showed that the transformation efficiency increased 100-fold only in the m6a$^+$ 2394 (Fig. 3). This led to a puzzling situation in which five of the seven strains had seemingly intact but inactive type I RM systems. One interpretation of this data is that RM systems could be a part of a larger network of defense systems in *E. faecium*. It is possible that there are multiple DNA defense elements, such as ArdA, working together simultaneously, which is why *E. faecium* is so difficult to genetically manipulate. It is also possible that the laboratory methods that we used to make *E. faecium* electrocompetent were not optimal.

ArdA anti-restriction proteins are typically found on mobile genetic elements such as plasmids and transposons (18, 34). From the 805 *E. faecium* strains analyzed in this study, 76.7% were shown to contain at least one *ardA* gene, with four different ArdA alleles identified (Table S6). The distribution of ArdA alleles indicates a potential avenue for overcoming type I RM systems and allowing for horizontal gene transfer between strains containing different HsdS alleles and different recognition sequences. However, when the two *ardA* genes present in *E. faecium* strain AUS0233 (*ardA1* and *ardA2*) were deleted, we did not observe any change in the function of the type I system. This suggests that additional components are present in AUS0233 that impact the methylase activity of the HsdM$_2$HsdS$_1$ complex. Through heterologous expression in *S. aureus,* we could show that both AUS0233 ArdA proteins were functional, leading to the complete knockdown of the type I restriction, allowing for maximal transformation efficiency with unmethylated DNA.

Overall, we observed that there is substantial diversity in the HsdS alleles present in *E. faecium*, but functional analysis suggests that allele presence does not always correlate with activity. Four dominant HsdS alleles are present in *E. faecium,* with clade A1 strains more likely to carry type I RM (73.7% of clade A1 strains carry one HsdS allele compared to 12%–13% of clade B or A2 strains). This could be the result of the coevolutionary "arms race" between bacteria and bacteriophage, causing the generation of diverse HsdS alleles and recognition motifs (38). This HsdS diversity between clades could lead to further niche adaptation and the generation of subpopulations. The lack of RM system functionality in many strains with low transformation efficiency suggests that additional DNA defense mechanisms could be operating in *E. faecium* that are yet to be identified.

## MATERIALS AND METHODS

### Media and reagents

*E. faecium* strains used in this study were routinely grown statically at 37°C in BHI (brain heart infusion) (Bacto, Difco) unless stated otherwise. *E. coli* and *S. aureus* were grown at 37°C with 200 rpm shaking, in Luria broth (LB) and BHI, respectively. All strains, plasmids, and primers are described in Tables 1 and 2. The addition of antibiotics and compounds to media was conducted at the following concentrations: chloramphenicol 10 µg/mL (Sigma), glycine (1%–3% wt/vol) (Chem Supply), and X-gal 100 µg/mL (Meridian Bioscience) for *E. faecium* and chloramphenicol 10 µg/mL, X-gal 100 µg/mL, ampicillin 100 µg/mL (Fisher Bioreagents), and kanamycin 50 µg/mL (Acros Organics) for *E. coli*. For *S. aureus,* chloramphenicol 10 µg/mL and kanamycin 50 µg/mL were used.

### Transformation of *E. coli*

Competent *E. coli* strains were generated by the rubidium chloride method and transformed by heat shock (42).

**TABLE 1** Bacteria and plasmids used in this study

| Strain | MDU-PHL strain ID | Species | Source |
|---|---|---|---|
| AUS0233 | | *E. faecium* ST796 | (28) |
| 2400 | AUSMDU00034400 | *E. faecium* ST1421 | MDU-PHL |
| 2399 | AUSMDU00023674 | *E. faecium* ST 203 | MDU-PHL |
| 2407 | AUSMDU00023358 | *E. faecium* ST 1421 | MDU-PHL |
| 2392 | AUSMDU00023361 | *E. faecium* ST 80 | MDU-PHL |
| 2391 | AUSMDU00023070 | *E. faecium* ST 80 | MDU-PHL |
| 2405 | AUSMDU00023341 | *E. faecium* ST 1421 | MDU-PHL |
| 2406 | AUSMDU00023351 | *E. faecium* ST 1421 | MDU-PHL |
| 2404 | AUSMDU00023338 | *E. faecium* ST 1421 | MDU-PHL |
| 2403 | AUSMDU00022892 | *E. faecium* ST 1421 | MDU-PHL |
| 2401 | AUSMDU00022752 | *E. faecium* ST 1421 | MDU-PHL |
| 2402 | AUSMDU00022764 | *E. faecium* ST 1421 | MDU-PHL |
| 2397 | AUSMDU00022795 | *E. faecium* ST 203 | MDU-PHL |
| 2398 | AUSMDU00022212 | *E. faecium* ST 80 | MDU-PHL |
| 2395 | AUSMDU00052092 | *E. faecium* ST 203 | MDU-PHL |
| 2396 | AUSMDU00052117 | *E. faecium* ST 203 | MDU-PHL |
| 2393 | AUSMDU00052483 | *E. faecium* ST 203 | MDU-PHL |
| 2394 | AUSMDU00051975 | *E. faecium* ST 203 | MDU-PHL |
| 2389 | AUSMDU00022797 | *E. faecium* ST 203 | MDU-PHL |
| 2390 | AUSMDU00022224 | *E. faecium* ST 80 | MDU-PHL |
| JE2 | | *S. aureus* | (40) |
| V583 | | *E. faecalis* | (41) |
| IM08B | | *E. coli, hsdMS* genes from *S. aureus* clonal complex 8, Δ*dcm* | (42) |
| DC10B | | *E. coli,* DH10BΔ*dcm* | (43) |
| AUS0233Δ*hsdMSR* | | *E. faecium* | This paper |
| 2406Δ*hsdR* | | *E. faecium* | This paper |
| 2394Δ*hsdR* | | *E. faecium* | This paper |
| **Plasmid** | | **Description** | **Source** |
| pIMAY-Z | | 8,815 bp, p15A, *lacZ*, G+ve *repA(ts)*, Cat$^R$ | (42) |
| pIMAY-Z(*hsdR*_2406) | | 9,951 bp, SOE insert 1,232 bp, p15A, *lacZ*, G+ve *repA(ts)*, Cat$^R$ | This paper |
| pIMAY-Z(*hsdR*_2394) | | 9,850 bp, SOE insert 1,131 bp, p15A, *lacZ*, G+ve *repA(ts)*, Cat$^R$ | This paper |
| pIMAY-Z(*hsdMSR)* | | 9,866 bp, SOE insert 1,147 bp, p15A, *lacZ*, G+ve *repA(ts)*, Cat$^R$ | This paper |
| pIMAY-Z(*ardA1*) | | 9,827 bp, SOE insert 1,108 bp, p15A, *lacZ*, G+ve *repA(ts)*, Cat$^R$ | This paper |
| pIMAY-Z(*ardA2*) | | 9,810 bp, SOE insert 1,091 bp, p15A, *lacZ*, G+ve *repA(ts)*, Cat$^R$ | This paper |
| pIMC8(P*gap*-YFP) | | 5,492 bp, p15A, *repA*, Cat$^R$, P*gap*-YFP | (44) |
| pRAB11*RBS | | 6,463 bp, tet inducible, Cat$^R$(G+ve), Amp$^R$(G−ve), pC194 (G+ve rep) | (45) |
| pRAB11(*ardA1*) | | 6,955 bp, Aus0233 anti-restriction protein 1 | This study |
| pRAB11(*ardA2*) | | 6,961 bp, Aus0233 anti-restriction protein 2 | This study |
| pCN34*RBS | | 6,401 bp Kan$^R$ (G+ve), Amp$^R$ (G−ve), pT181 rep (G+ve) | (45) |

## *E. faecium* electrocompetent cells

A 10 mL BHI broth was inoculated with a single *E. faecium* colony and grown overnight at 37°C. Then, a 50 mL BHI broth containing 500 mM sucrose and the optimal glycine concentration was inoculated with 500 µL of the previous overnight culture (the optimal glycine concentration inhibited growth by 50% compared to BHI only) and grown at 37°C overnight. Cells were harvested at room temperature for 10 minutes at 5,000 × *g* with the supernatant discarded. The cells were resuspended in 50 mL of prewarmed fresh BHI broth containing 500 mM sucrose and the optimal glycine concentration and incubated at 37°C for 1 hour. The cells were harvested at 4°C for 10 minutes at 5,000 × *g* and subsequently resuspended in 25 mL of ice-cold wash buffer (10% glycerol, 500 mM sucrose, pH 7.0, filter sterilized) for three separate washes. Finally, the cells were

**TABLE 2** Oligonucleotides used in this study

| Primer name | Primer sequence (5′–3′) | Target | Strain | Source |
|---|---|---|---|---|
| AK5(A) | CCTCACTAAAGGGAACAAAAGCTGGGTACC AAGATTATTCTGTGGTATTTAAGAACGGAC | *hsdMSR* | AUS0233 | This paper |
| AK6(B) | CAAAAAGATACACTCCTATCTCTTATTTAAAG TTGC | *hsdMSR* | AUS0233 | This paper |
| AK7(C) | AAATAAGAGATAGGAGTGTATCTTTTTGTAAA CGAGAAGAGTTAAGAAAATTGTACG | *hsdMSR* | AUS0233 | This paper |
| AK8(D) | CGACTCACTATAGGGCGAATTGGAGCTCCATCG AATATCATTTAAAATCAATACGACG | *hsdMSR* | AUS0233 | This paper |
| AK13(A) | CCTCACTAAAGGGAACAAAAGCTGGGTACCAAG GTGGTTCTGAAGGTACTC | *hsdR* | 2394 | This paper |
| AK14(B) | CATTCTTCTTCCCTCCTAGAC | *hsdR* | 2394 | This paper |
| AK15(C) | GTCTAGGAGGGAAGAAGAATGTAAACGAGAAG AGTTAAGAAAATTGTACG | *hsdR* | 2394 | This paper |
| AK16(D) | CGACTCACTATAGGGCGAATTGGAGCTCCAAACT GAGTCATATAGAGACAAAAATTTC | *hsdR* | 2394 | This paper |
| AK23(A) | CCTCACTAAAGGGAACAAAAGCTGGGTACCGATT ATTCCCTTGTTCGATATAGACC | *hsdR* | 2406 | This paper |
| AK24(B) | TAAACGAGAAGAGTTAAGAAAATTGTACG | *hsdR* | 2406 | This paper |
| AK25(C) | CGTACAATTTTCTTAACTCTTCTCCATTCTTCTTCC CTCCTAGAC | *hsdR* | 2406 | This paper |
| AK26(D) | CGACTCACTATAGGGCGAATTGGAGCTCTTGGGA GAATTAGGTAAAACTCAATCG | *hsdR* | 2406 | This paper |
| IM1 | GGTACCCAGCTTTTGTTCCCTTTAGTGAGG | pIMAY-Z | | (42) |
| IM2 | GAGCTCCAATTCGCCCTATAGTGAGTCG | pIMAY-Z | | (42) |
| IM515 | TTTCCAATTCCTCCTCATCATACTCTATC | pRAB11*RBS | | (45) |
| IM1356 | GTTAACAGATCTGAGCTCGAATTCACTGG | pRAB11*RBS | | (45) |
| AK72(A) | CCTCACTAAAGGGAACAAAAGCTGGGTACCGGAAAAG TTTGAGATATTTATAATATCAGG | *ardA1* | AUS0233 | This paper |
| AK73(B) | CATATATTCACGTCCTTTCTTTGTAG | *ardA1* | AUS0233 | This paper |
| AK74(C) | CTACAAAGAAAGGACGTGAATATATGTAAATCTGTCGG TACATTACTAC | *ardA1* | AUS0233 | This paper |
| AK75(D) | CGACTCACTATAGGGCGAATTGGAGCTCGATTCTGTA CTGATTTGTAAAGC | *ardA1* | AUS0233 | This paper |
| AK76(A) | CCTCACTAAAGGGAACAAAAGCTGGGTACCACATTGT CATCATATTGTACCAAC | *ardA2* | AUS0233 | This paper |
| AK77(B) | CATCGTGCTTCCCCCTTAC | *ardA2* | AUS0233 | This paper |
| AK78(C) | GTAAGGGGGAAGCACGATGTAACAAAATTATTGAAG GGTAGTTG | *ardA2* | AUS0233 | This paper |
| AK79(D) | CGACTCACTATAGGGCGAATTGGAGCTCAAGGAGTAA TTGTAAGGTATTAACTCATAG | *ardA2* | AUS0233 | This paper |
| AK92 | GATAGAGTATGATGAGGAGGAATTGGAAA ATGGACGATATGCAAGTCTATATTG | *ardA1* | AUS0233 | This paper |
| AK93 | CCAGTGAATTCGAGCTCAGATCTGTTAACTTAATAGACG ATTTCAAAAATCCCATGATTGG | *ardA1* | AUS0233 | This paper |
| AK98 | GATAGAGTATGATGAGGAGGAATTGGAAAATGGAAC AGATGCGTGTTTATATTG | *ardA2* | AUS0233 | This paper |
| AK99 | CCAGTGAATTCGAGCTCAGATCTGTTAACTTAATAT GGATACTCAAAAACACCTC | *ardA2* | AUS0233 | This paper |

resuspended in 500 µL of wash buffer, and 5 × 100 µL aliquots were dispensed into pre-chilled Eppendorf tubes and frozen at −80°C (stored 1 hour before electroporation, if on the same day).

## Electroporation of *E. faecium*

Competent cells were thawed on ice for 5 minutes. A 1 µg aliquot of plasmid was added to the competent cells, mixed, and then kept on ice for 5 minutes. The mixture was then added to a 0.2 cm electroporation cuvette (Bio-Rad) chilled on ice. The cells were electroporated (Gene Pulser Xcell, Bio-Rad) at 2.5 kV, 25 µF, and 200 Ω. Following electroporation, 1 mL of ice-cold BHI containing 500 mM sucrose was added, and the cells were incubated at 30℃ or 37℃ for 2 hours, statically. Cells were then diluted and plated onto BHI agar containing 10 µg/mL of chloramphenicol (Cm) and incubated for 24–72 hours.

## Generation of mutants in *E. faecium* using allelic exchange

Markerless deletions of genes in *E. faecium* were constructed using the allelic exchange plasmid pIMAY-Z, as described previously (42, 46). Briefly, upstream (AB) and downstream (CD) regions of the target gene were amplified using standard Phusion (NEB) PCR conditions. To amplify the upstream and downstream flanks, the A/B and C/D primer pairs were used, respectively, for each strain. These primer sets were used to generate the following mutants: Δ*hsdMSR* in AUS0233 [AK5(A)/AK6(B); AK7(C)/AK8(D)]; Δ*hsdR* in 2394 [AK13(A)/AK14(B); AK15(C)/AK16(D)]; Δ*hsdR* in 2406 [AK23(A)/AK24(B); AK25(C)/AK26(D)]. These primer sets were used to generate the following mutants: Δ*ardA1* in AUS0233 [AK72(A)/AK73(B); AK74(C)/AK75(D)]; Δ*ardA2* in AUS0233 [AK76(A)/AK77(B); AK78(C)/AK78(D)]. The AB and CD products were joined by spliced overlap extension-PCR (SOE-PCR) and SLiCE cloned into pIMAY-Z as previously described in Monk and Stinear (42). Some transformed *E. faecium* colonies took 48–72 hours to yield blue colonies at 30℃ post-transformation. All mutants were whole-genome sequenced by Illumina sequencing on the Illumina NextSeq 550 platform with 2 × 150 bp paired-end reads as per the manufacturer's instructions by the Centre of Pathogen Genomics Innovation Hub (CPG-IH, University of Melbourne). DNA extraction was carried out using the JANUS automated workstation (PerkinElmer) and Chemagic magnetic bead technology (PerkinElmer). DNA libraries were prepared using the Nextera XT kit according to the manufacturer's instructions (Illumina Inc.). The resulting Illumina reads were mapped to the reference genome using Geneious mapper to confirm the accuracy of the mutation by visualizing the loss of reads mapping to the target gene in the reference and that there were no secondary mutations acquired using Geneious Prime (v2023) (https://www.geneious.com/).

## Genomic DNA extraction

Genomic DNA was extracted from 2 mL of an overnight culture using the New England Biolabs (NEB) Monarch Genomic DNA Purification kit. The cell pellet was washed with phosphate-buffered saline (PBS), resuspended in 100 µL of lysis buffer (PBS containing 30 µL of lysozyme [100 mg/mL stock] and 10 µL of RNaseA [10 mg/mL stock]), and then 100 µL of tissue lysis buffer was added. The cells were incubated at 37℃ for 1 hour before the addition of 10 µL of Proteinase K (20 mg/mL stock). The cells were then incubated at 56℃ for 30 minutes at 1,400 rpm and then processed following the manufacturer's instructions.

## Construction of pRAB11(ardA1) or pRAB11(ardA2)

We cloned *ardA1* or *ardA2* into the tetracycline-inducible expression vector pRAB11*RBS to construct pRAB11(*ardA1*) or pRAB11(*ardA2*) (45). The plasmid pRAB11*RBS was linearized using the restriction enzyme KpnI, gel extracted, and PCR amplified using primers IM515/IM1356. The *ardA1* (primers AK92 and AK93) or *ardA2* (primers AK98 and AK99) genes were amplified from AUS0233 genomic DNA. The PCR products were gel extracted (Monarch DNA Gel Extraction kit, NEB), SLiCE cloned into pRAB11*RBS, and transformed into *E. coli* IM08B and selected for with 100 µg/mL of ampicillin on L-agar

(42). The insert confirmed that the plasmid was isolated with the Monarch Plasmid Miniprep kit (NEB), sequenced on the ONT MinION sequencing platform (CPG-IH), and assembled using Raven (47, 48).

## Testing the impact of ArdA1 on the transformation of *S. aureus* JE2

*S. aureus* electrocompetent cells were generated using a method adapted from Monk and Stinear (42). A 10 mL overnight BHI culture containing Cm 10 µg/mL [JE2+pRAB11, JE2+pRAB11(*ardA1*), or JE2+pRAB11(*ardA2*)] was diluted to 0.5 $OD_{600}$ into 100 mL of prewarmed BHI (250 mL flask) containing 100 ng/mL of aTc and 10 µg/mL of Cm. Cells were grown to an $OD_{600}$ of 1.0 and then harvested at 7,000 × *g* for 5 minutes at 4°C. Cells were then washed twice with ice-cold water and twice with ice-cold 10% (wt/vol) glycerol. The cells were finally resuspended in 500 µL and dispensed in 10 × 50 µL aliquots and stored at −80°C. Competent cells were then used to test the transformation efficiency with the pCN34*RBS plasmid extracted from either DC10B or IM08B (45). Electroporation was performed as described previously (42), with the transformants selected on BHI agar containing 50 µg/mL of kanamycin.

## Genome assembly of test panel isolates

The test panel of *E. faecium* isolates was obtained from the Microbiological Diagnostic Unit-Public Health Laboratory (MDU-PHL) at the University of Melbourne. DNA library prep was conducted using the NexteraXT DNA preparation kit and sequenced on the Illumina NextSeq 500 platform (28). Reads for the test panel strains were *de novo* assembled using SPAdes genome assembler (v3.15.2) to generate contigs (28, 49). These contigs were annotated using Prokka (v.1.14.6) (https://github.com/tseemann/prokka).

## Trycycler assembly of PacBio reads

Genomic DNA was isolated as described previously from the following strains: AUS0233, 2391, 2394, 2395, 2396, 2397, 2405, and 2406. DNA was size-selected with the short-read eliminator kit (Circulomics). A SMRT Bell library (SMRTbell express template prep kit 2.0, PacBio) was made for each strain, barcoded with the Barcoded Overhang Adapter Kit 8A (PacBio), and loaded into Sequel (PacBio). The PacBio long reads were assembled using Trycycler (https://github.com/rrwick/Trycycler) (v.0.5.5) following the "extra-thorough assembly" instructions in the documentation and then polished using Illumina short reads (above) with Polypolish (v.0.6.0) (https://github.com/rrwick/Polypolish) and Pypolca (v0.3.1) (https://github.com/gbouras13/pypolca) (50). The PacBio Trycycler assembled genomes were DnaA corrected using the tool Circulator (v1.5.5) (https://github.com/sanger-pathogens/circlator), and the genomes were annotated using Prokka (v.1.14.6) (https://github.com/tseemann/prokka) (51, 52). Methylome calling used the genome assembly files on the SMRTlink portal (https://github.com/WenchaoLin/SMRT-Link). SMRT analysis was used on each genome FASTA file. The closed genome sequence was analyzed for modified base motifs. Then, base consensus calling was conducted to identify m6A methylation. The "pairs of motifs" and matching recognition sequences were analyzed. Recognition sequences for m6A methylation and frequency of methylation on each site across the genome were recorded (v0.3.1) (50). The PacBio Trycycler assembled genomes were DnaA using the tool Circulator (v1.5.5) (https://github.com/sanger-pathogens/circlator), and the genomes were annotated using Prokka (v.1.14.6) (https://github.com/tseemann/prokka) (51, 52). Methylome calling used the genome assembly files on the SMRTlink portal (https://github.com/WenchaoLin/SMRT-Link). SMRT analysis was used on each genome FASTA file. The closed genome sequence was analyzed for modified base motifs. Then, base consensus calling was conducted to identify m6A methylation. The "pairs of motifs" and matching recognition sequences were analyzed. Recognition sequences for m6A methylation and frequency of methylation on each site across the genome were recorded.

## *In silico* DNA defense system analysis

"Defense Finder" (v1.0.9) was used on SPAdes assembled Illumina contigs as per instructions on the GitHub (https://github.com/mdmparis/defense-finder) (20, 21). Blast (v0.10.0) (https://github.com/mosuka/blast) was used as per the GitHub instructions to search for gene presence.

## Sequence alignment and visualization

Sequences and alignments were visualized using Geneious Prime. DNA and protein sequences were aligned using Clustal Omega (53). Amino acid alignment trees were generated using Geneious Tree Building using the Blosum62 cost matrix and default tree builder options. Long-read DNA sequences were aligned to reference genomes using Minimap2, short-read DNA sequences were aligned to reference genomes using the Geneious mapper tool (Geneious Prime [v2023]), and plasmid sequences longer than 5 kb were aligned using LASTZ (53–55).

## Global strain collection acquisition

Raw Illumina reads for the 785 global *E. faecium* strains from van Hal et al. (10) were downloaded from the SRA archive from the 69 different project numbers found in Table S1 (10). Draft genome assemblies were generated from raw reads downloaded from SRA using shovill v1.1.0 (https://github.com/tseemann/shovill) with the SPAdes genome assembler (v3.15.2) (49). Quality control was conducted on the assembled genomes by assessing the number of contigs, overall base pair number, and minimum contig size per genome to ensure they were the expected size for an *E. faecium* genome. Strains that were duplicates were removed from this study.

## Phylogenetic analysis

A core SNP alignment was generated with Snippy (v4.4.5) (https://github.com/tseemann/snippy) with a phylogenetic tree using FastTree (v1.6.0) using the AUS0085 reference genome (56, 57). This tree was then designed using Interactive Tree Of Life (iTOL) (58). In the 805 strain phylogenetic tree, recombination was filtered using ClonalFrameML (v1.13) to mask the alignment so that it is 95% core rather than a strict 100% core SNP alignment (59). This maximum likelihood tree was also bootstrapped (0.7) to increase branch confidence using FastTree (60).

## Global *E. faecium* HsdS and ArdA analysis

BlastX was used to screen a collection of 805 draft genome assemblies for the eight alleles obtained from the following repository (https://gitlab.com/sirarre-dondo/efaecium_population/-/blob/master/Files/rmsystem/rm_collection.csv), using a cut-off threshold of 100% identity and full length of amino acids for each HsdS allele (12). We used a 40aa length of each HsdS allele as bait for the blast search and then ordered the BlastX hits by the full-length cut-off threshold. The length for the main eight HsdS alleles was HsdS_1 (WP_002287733.1) 414aa, HsdS_2 (WP_002330870.1) 405aa, HsdS_3 (WP_002347317.1) 399aa, HsdS_4 (WP_242618483.1) 405aa, HsdS_5 (WP_002332926.1) 412aa, HsdS_6 (WP_002324520.1) 409aa, HsdS_7 (WP_126309379.1) 379aa, and HsdS_8 (WP_002341679.1) 403aa. For the ArdA blastX search, a cut-off of 100% identity and alignment coverage of the amino acid length for each allele (ArdA1_0233_H2 [WP_000342539.1]: 165aa, ArdA2_0233 [WP_002297350.1]: 167aa, 2405_H1 [WP_084959603.1]: 166aa, and Efm_Uniprot_Q3XYD3 [WP_002288934.1]: 166aa) was used to identify ArdA protein carriage in the 805 global strains (61).

## Panaroo gene presence-absence analysis

The pangenome clustering tool Panaroo (v1.5.1) was used to analyze the Trycycler-assembled genomes for the presence and absence of genes between genomes and plasmid sequence using .gff files (62), created with Prokka (52).

## ACKNOWLEDGMENTS

We would like to thank MDU-PHL for suppling *E. faecium* isolates used in the study.

## AUTHOR AFFILIATIONS

[1]Department of Microbiology and Immunology, The University of Melbourne, at the Peter Doherty Institute for Infection and Immunity, Melbourne, Victoria, Australia
[2]Centre for Pathogen Genomics - Innovation Hub, The University of Melbourne, Melbourne, Australia

## AUTHOR ORCIDs

Timothy P. Stinear http://orcid.org/0000-0003-0150-123X
Andrew H. Buultjens http://orcid.org/0000-0002-5984-1328
Ian R. Monk http://orcid.org/0000-0001-6982-8074

## FUNDING

| Funder | Grant(s) | Author(s) |
|---|---|---|
| National Health and Medical Research Council | GNT1194325 | Timothy P. Stinear |

## AUTHOR CONTRIBUTIONS

Alexandra L. Krause, Formal analysis, Investigation, Methodology, Writing – original draft, Writing – review and editing | Louise M. Judd, Data curation, Formal analysis, Methodology, Resources, Writing – review and editing | Ryan Wick, Formal analysis, Methodology, Software, Writing – review and editing | Timothy P. Stinear, Conceptualization, Data curation, Formal analysis, Funding acquisition, Investigation, Methodology, Resources, Supervision, Validation, Visualization, Writing – original draft, Writing – review and editing | Andrew H. Buultjens, Data curation, Formal analysis, Investigation, Methodology, Software, Supervision, Visualization, Writing – review and editing | Ian R. Monk, Conceptualization, Formal analysis, Investigation, Methodology, Resources, Supervision, Writing – review and editing

## DATA AVAILABILITY

The nanopore and Illumina reads for the study have been deposited at NCBI under the accession number PRJNA1223019.

## ADDITIONAL FILES

The following material is available online.

### Supplemental Material

**Supplemental material (Spectrum00289-25-s0001.pdf).** Fig. S1 to S5; Tables S2, S4, S5, and S7.
**Table S1 (Spectrum00289-25-s0002.xlsx).** Global *E. faecium* metadata from the 805 strains.
**Table S3 (Spectrum00289-25-s0003.xlsx).** Gene presence-absence analysis of *E. faecium* strains.
**Table S6 (Spectrum00289-25-s0004.xlsx).** HsdS and ArdA allele carriage across the different MLST types.

## Open Peer Review

**PEER REVIEW HISTORY (review-history.pdf).** An accounting of the reviewer comments and feedback.

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
