## [Reviewer comments · Microbiology Spectrum]

Microbiology Spectrum

Comparative and functional genomic analysis of foreign DNA defence mechanisms in *Enterococcus faecium*

Alexandra Krause, Louise Judd, Ryan Wick, Timothy Stinear, Andrew Buultjens, and Ian Monk

Corresponding Author(s): Ian Monk, The University of Melbourne

Review Timeline:

Submission Date:	January 30, 2025
Editorial Decision:	April 24, 2025
Revision Received:	May 19, 2025
Accepted:	May 26, 2025

Editor: Daria Van Tyne

Reviewer(s): Disclosure of reviewer identity is with reference to reviewer comments included in decision letter(s). The following individuals involved in review of your submission have agreed to reveal their identity: Willem Van Schaik (Reviewer #1)

Transaction Report:

DOI: <https://doi.org/10.1128/spectrum.00289-25>

Re: Spectrum00289-25 (Comparative and functional genomic analysis of foreign DNA defence mechanisms in *Enterococcus faecium*)

Dear Dr. Ian Robertson Monk:

Thank you for the privilege of reviewing your work. Your manuscript has been reviewed by two experts, and I would now like you to revise your study in line with their feedback. Below you will find instructions from the Spectrum editorial office and the reviewer comments.

Revision Guidelines

Sincerely,
Daria Van Tyne
Editor
Microbiology Spectrum

Reviewer #1 (Comments for the Author):

The manuscript describes a substantial amount of work that has been completed to a high standard. The conclusions are supported by the data provided in the manuscript, but in a few cases data could be presented more clearly and more background information on restriction modification systems, particularly the role of the HsdS and ArdA proteins, should be provided in the introduction to make this manuscript accessible to a larger audience.

Specific comments:

The genome sequence data does not appear to be available in a sequence repository. The data needs to be uploaded to NCBI/ENA and be made available for re-use and re-analysis.

The introduction does not describe the exact role of HsdS as a specificity subunit of restriction modification systems. A more in-depth introduction to HsdS needs to be included in the introduction to aid the reader. Otherwise it will not be clear why HsdS is discussed in significant detail in the rest of the manuscript. Similarly, ArdA needs to be shortly introduced in the introduction as well as this would help the reader understand why the authors switch their attention to this protein in the course of the manuscript.

I. 26. In line with the comment above, include the function of HsdS here.

I. 34 - 36. This section seems to highlight the data on the *ardA* mutant, but would it also not be of interest to mention the effect of *hsdMSR* and *hsdR* deletions described in line 210 - 216 and Fig. 3?

I. 47 - 48. Similarly, this line does appear to reflect the data in I. 210 - 216.

I. 61. Remove stray full stop.

I. 72 - 85: this is the section where additional text on the function and mechanism of HsdS and ArdA can be described.

I. 130 - 131. It is not immediately obvious why the presence of an adenine methylating type IIG system and type I system would be incompatible: this may again be due to the lack of detail in the introduction.

I. 183: I believe 'of' in 'of which' can be deleted.

I. 227: review the bold formatting (does it signify anything?) and remove the stray parenthesis.

I. 238: write 'the HsdS_2 allele'

I. 269: why is reference 30 cited here? Perhaps it is better to include this in the methods section?

I. 295 - 299: this section is difficult to parse. The authors write that there is a correlation between the presence of HsdS and ArdA with 382 strains having at least one ArdA and HsdS allele. However a similar number ($225 + 134 = 359$) only have a HsdS or ArdA allele, so it is not clear whether it can be suggested that 'a correlation between the presence of these alleles' can be suggested.

I. 306: it is probably better to write 'similar, but not identical, to Tn916.'

References: many references lack page numbers or article identifiers.

Reviewer #2 (Comments for the Author):

In this work, the authors examine barriers to horizontal gene transfer in hospital-adapted *Enterococcus faecium* strains, particularly the effect of Type I restriction-modification systems and anti-restriction ArdA proteins. 805 strains were examined computationally; the transformation efficiency of 20 of these was examined experimentally; 8 of these were sequenced and their methylomes examined using the PacBio platform. Most of the 805 strains contained at least 1 Type I RM system, but methylome analysis suggested that the majority of these may be silenced. Most of the 805 strains also contained at least 1 *ardA* allele, suggesting a possible method for the silencing, but deletion experiments showed that, by and large, they are not, at least not in *E. faecium*. The mechanism of RM suppression remains unknown.

Despite not identifying the mechanism of suppression, this work adds to our knowledge of the overall pictures of transformation barriers and genome defense mechanisms in *E. faecium*. I have a few minor comments and questions.

On the one hand, one might expect fewer barriers to HGT in hospital adapted strains (A1) than in the broader environment (A2 and B), given the need for acquisition of antibiotic-resistance factors. The greater number of plasmids in A1 suggests this as well. Yet, we see a preponderance of Type I RM systems (silent or not) in A1, even excluding the plasmid-associated HsdS_4. Do the authors have an explanation? Are hospital-adapted strains exposed to more frequent phage attacks?

There seems to be little correlation between Type I system activity and transformation efficiency, at least in the set of 8 sequenced strains. (3/8 strains show m6A methylation, 2/20 strains show poor transformation, and only one strain shows both

phenotypes.) In that one case it is hsdR that is responsible [Fig 3], but methylation overall is poor, suggesting a possible phase-variable repressor. Perhaps looking for hallmarks of phase variable mechanisms (repeats, etc.) in the sequenced genome might help identify other factors at play here.

In Figure 1, I find the "orange" and "light orange" very difficult to distinguish. Can they be made more different?

Lines 130-131. Although incompatibility can't be ruled out, it is hard to make that case based on one example.

Line 211. Why was strain 2397 not used as the "control" for 2394, since its HsdS_3 is stated to be 100% identical to 2394 (lines 169-70, Fig S1), and it also does not have the potentially confounding second Type I system, HsdS_4?

In Supp Fig S3, the significance of the colored circles should be noted. Are these HsdS_1-4? Also, it might be helpful to add the HsdS allele numbers as well as representative recognition sites, where known.

Lines 270-1. The authors seem to be arguing that the lack of hsdS alleles in non-clinical isolates is due to lower selection pressure to acquire MGEs in their environment. The converse of this would be the greater number of hsdS alleles in the hospital environment is due to greater selection pressure to acquire MGEs, which again seems counterintuitive to me, since RM systems are generally thought of as barriers to HGT. Can this be clarified?

Line 298-9. Of 681 A1 strains, 86% have an ardA allele (~586). Of these, 225 lack an HsdS (38%). This does not seem like much of a correlation.

There seems to be significant recombination between the hsdS alleles, or at least the TRDs within (Supp Fig S3). What about the hsdR and M genes? Are these largely identical between the systems?

[This is the same as what's in "comments and suggestions for the authors".]

In this work, the authors examine barriers to horizontal gene transfer in hospital-adapted *Enterococcus faecium* strains, particularly the effect of Type I restriction-modification systems and anti-restriction ArdA proteins. 805 strains were examined computationally; the transformation efficiency of 20 of these was examined experimentally; 8 of these were sequenced and their methylomes examined using the PacBio platform. Most of the 805 strains contained at least 1 Type I RM system, but methylome analysis suggested that the majority of these may be silenced. Most of the 805 strains also contained at least 1 *ardA* allele, suggesting a possible method for the silencing, but deletion experiments showed that, by and large, they are not, at least not in *E. faecium*. The mechanism of RM suppression remains unknown.

Despite not identifying the mechanism of suppression, this work adds to our knowledge of the overall pictures of transformation barriers and genome defense mechanisms in *E. faecium*. I have a few minor comments and questions.

On the one hand, one might expect fewer barriers to HGT in hospital adapted strains (A1) than in the broader environment (A2 and B), given the need for acquisition of antibiotic-resistance factors. The greater number of plasmids in A1 suggests this as well. Yet, we see a preponderance of Type I RM systems (silent or not) in A1, even excluding the plasmid-associated *HsdS_4*. Do the authors have an explanation? Are hospital-adapted strains exposed to more frequent phage attacks?

There seems to be little correlation between Type I system activity and transformation efficiency, at least in the set of 8 sequenced strains. (3/8 strains show m6A methylation, 2/20 strains show poor transformation, and only one strain shows both phenotypes.) In that one case it is *hsdR* that is responsible [Fig 3], but methylation overall is poor, suggesting a possible phase-variable repressor. Perhaps looking for hallmarks of phase variable mechanisms (repeats, etc.) in the sequenced genome might help identify other factors at play here.

In Figure 1, I find the "orange" and "light orange" very difficult to distinguish. Can they be made more different?

Lines 130-131. Although incompatibility can't be ruled out, it is hard to make that case based on one example.

Line 211. Why was strain 2397 not used as the "control" for 2394, since its HsdS_3 is stated to be 100% identical to 2394 (lines 169-70, Fig S1), and it also does not have the potentially confounding second Type I system, HsdS_4?

In Supp Fig S3, the significance of the colored circles should be noted. Are these HsdS_1-4? Also, it might be helpful to add the HsdS allele numbers as well as representative recognition sites, where known.

Lines 270-1. The authors seem to be arguing that the lack of hsdS alleles in non-clinical isolates is due to lower selection pressure to acquire MGEs in their environment. The converse of this would be the greater number of hsdS alleles in the hospital environment is due to greater selection pressure to acquire MGEs, which again seems counterintuitive to me, since RM systems are generally thought of as barriers to HGT. Can this be clarified?

Line 298-9. Of 681 A1 strains, 86% have an ardA allele (~586). Of these, 225 lack an HsdS (38%). This does not seem like much of a correlation.

There seems to be significant recombination between the hsdS alleles, or at least the TRDs within (Supp Fig S3). What about the hsdR and M genes? Are these largely identical between the systems?

[This is the same as what's in "comments and suggestions for the authors".]

In this work, the authors examine barriers to horizontal gene transfer in hospital-adapted *Enterococcus faecium* strains, particularly the effect of Type I restriction-modification systems and anti-restriction ArdA proteins. 805 strains were examined computationally; the transformation efficiency of 20 of these was examined experimentally; 8 of these were sequenced and their methylomes examined using the PacBio platform. Most of the 805 strains contained at least 1 Type I RM system, but methylome analysis suggested that the majority of these may be silenced. Most of the 805 strains also contained at least 1 *ardA* allele, suggesting a possible method for the silencing, but deletion experiments showed that, by and large, they are not, at least not in *E. faecium*. The mechanism of RM suppression remains unknown.

Despite not identifying the mechanism of suppression, this work adds to our knowledge of the overall pictures of transformation barriers and genome defense mechanisms in *E. faecium*. I have a few minor comments and questions.

On the one hand, one might expect fewer barriers to HGT in hospital adapted strains (A1) than in the broader environment (A2 and B), given the need for acquisition of antibiotic-resistance factors. The greater number of plasmids in A1 suggests this as well. Yet, we see a preponderance of Type I RM systems (silent or not) in A1, even excluding the plasmid-associated *HsdS_4*. Do the authors have an explanation? Are hospital-adapted strains exposed to more frequent phage attacks?

There seems to be little correlation between Type I system activity and transformation efficiency, at least in the set of 8 sequenced strains. (3/8 strains show m6A methylation, 2/20 strains show poor transformation, and only one strain shows both phenotypes.) In that one case it is *hsdR* that is responsible [Fig 3], but methylation overall is poor, suggesting a possible phase-variable repressor. Perhaps looking for hallmarks of phase variable mechanisms (repeats, etc.) in the sequenced genome might help identify other factors at play here.

In Figure 1, I find the "orange" and "light orange" very difficult to distinguish. Can they be made more different?

Lines 130-131. Although incompatibility can't be ruled out, it is hard to make that case based on one example.

Line 211. Why was strain 2397 not used as the "control" for 2394, since its HsdS_3 is stated to be 100% identical to 2394 (lines 169-70, Fig S1), and it also does not have the potentially confounding second Type I system, HsdS_4?

In Supp Fig S3, the significance of the colored circles should be noted. Are these HsdS_1-4? Also, it might be helpful to add the HsdS allele numbers as well as representative recognition sites, where known.

Lines 270-1. The authors seem to be arguing that the lack of hsdS alleles in non-clinical isolates is due to lower selection pressure to acquire MGEs in their environment. The converse of this would be the greater number of hsdS alleles in the hospital environment is due to greater selection pressure to acquire MGEs, which again seems counterintuitive to me, since RM systems are generally thought of as barriers to HGT. Can this be clarified?

Line 298-9. Of 681 A1 strains, 86% have an ardA allele (~586). Of these, 225 lack an HsdS (38%). This does not seem like much of a correlation.

There seems to be significant recombination between the hsdS alleles, or at least the TRDs within (Supp Fig S3). What about the hsdR and M genes? Are these largely identical between the systems?

Response to reviewer comments

We have carefully considered each of the issues raised by the reviewers. Please find our point-by-point responses below. Page and line numbers refer to the marked-up, revised manuscript. Additional text is highlighted in yellow.

Reviewer #1 (Comments for the Author):

The manuscript describes a substantial amount of work that has been completed to a high standard. The conclusions are supported by the data provided in the manuscript, but in a few cases data could be presented more clearly and more background information on restriction modification systems, particularly the role of the HsdS and ArdA proteins, should be provided in the introduction to make this manuscript accessible to a larger audience.

Specific comments:

The genome sequence data does not appear to be available in a sequence repository. The data needs to be uploaded to NCBI/ENA and be made available for re-use and re-analysis.

We have uploaded the raw reads to NCBI under the accession number: PRJNA1223019

Data availability

The nanopore and illumina reads for the study have been deposited at NCBI under the accession number PRJNA1223019.

(Page 24/25; Lines 714-716)

The introduction does not describe the exact role of HsdS as a specificity subunit of restriction modification systems. A more in-depth introduction to HsdS needs to be included in the introduction to aid the reader. Otherwise it will not be clear why HsdS is discussed in significant detail in the rest of the manuscript. Similarly, ArdA needs to be shortly introduced in the introduction as well as this would help the reader understand why the authors switch their attention to this protein in the course of the manuscript.

I. 26. In line with the comment above, include the function of HsdS here.

I. 34 - 36. This section seems to highlight the data on the ardA mutant, but would it also not be of interest to mention the effect of hsdMSR and hsdR deletions described in line 210 - 216 and Fig. 3?

I. 47 - 48. Similarly, this line does appear to reflect the data in I. 210 - 216.

We have modified the **abstract** to improve clarity, also changes were subsequently required to reduce the word count:

Enterococcus faecium is notoriously difficult to study genetically due to the poor understanding of barriers preventing foreign DNA uptake such as the proteins that modify Type I restriction modification (RM) system activity. Here, we compared *E. faecium* repertoires of the HsdS specificity subunit (dictate the DNA motif that is adenine methylated) from Type I RM systems among 805 globally reported *E. faecium* isolates. We showed there were eight distinct HsdS types, with four dominant variants that were also significantly enriched in the hospital associated clade A1 *E. faecium* lineage. Adenine methylome analysis of a subset of eight representative *E. faecium* strains revealed only two exhibited functional type I RM systems, with the activity corroborated by the construction of type I RM deletion mutants. To investigate this surprising finding, we assessed the contribution of the anti-restriction protein ArdA that specifically inhibits type I RM function. The *E. faecium* ST796 clinical isolate AUS0233 has one intact Type I RM system, no adenine methylation and two distinct *ardA* paralogs. When heterologously expressed in *Staphylococcus aureus* JE2, both *E. faecium* *ardA* variants were functional, each inhibiting the function of the two, type I RM systems in *S. aureus*. However, the deletion of one or both versions of *ardA* in *E. faecium* AUS0233 did not change the transformation efficiency with exogenous DNA, suggesting ArdA in *E. faecium* AUS0233 is not controlling type I RM. This study highlights the complexity of DNA defence mechanisms in *E. faecium* and suggests unidentified factors control acquisition of foreign DNA.

(Page 2; Lines 21-37)

The text for the Importance has been updated as follows:

From PacBio analysis of *E. faecium* strains, it was observed that the majority of *E. faecium*, do not adenine methylate DNA despite genome analysis indicating they have intact type I RM methylation systems. One explanation for this observation is that *E. faecium* produces anti-restriction factors such as ArdA which can inhibit type I RM systems. However, the deletion of both *ardA* alleles did not improve efficiency of DNA uptake. (Page 3; Line 59-61)

I. 61. Remove stray full stop.

Removed

I. 72 – 85: this is the section where additional text on the function and mechanism of HsdS and ArdA can be described.

Additional information has been added on HsdS and ArdA function

Type I RM systems are comprised of three protein subunits, termed host specificity of DNA (Hsd), specificity (S), modification (HsdM - methylase) protein and the restriction endonuclease (HsdR) (14). The HsdM and HsdS form a protein complex (HsdM₂HsdS₁) that adenine methylate host DNA. The sequence methylated is dictated by the two-target recognition domains (TRD) of the HsdS which recognise an asymmetric bipartite DNA sequence. The two HsdS TRD motifs comprise 3–4 base pairs, which contain the methylated adenine residue, separated by 4 to 9 non-specific base pairs. During DNA replication, the newly replicated strand is adenine methylated at the HsdS dictated TRD motif in the presence of hemi-methylated template. Two HsdR subunits combine with the HsdM₂HsdS₁ complex in the absence of specific adenine methylation pattern, transforming the system into a molecular motor bound to the DNA, introducing DNA breaks through collision with either a second bound RM complex, other proteins or DNA secondary structure. (Pages 4-5; Lines 89-101)

Other elements such as CBASS (cyclic-oligonucleotide-based anti-phage signalling systems), *Abi* (abortive infection) or *ArdA* (mimics the shape and structure of a short DNA sequence to inhibit type I RM activity) could play a role in genome defence through discrimination of foreign from host DNA (16-18). (Page 5; Lines 103-105)

We examined the global presence and function of HsdS alleles in select isolates. Further we defined the impact of the anti-restriction protein, *ArdA*, on type I RM system function by assessing transformation efficiency in *E. faecium* AUS0233 and *Staphylococcus aureus* JE2. (Page 5; Line 115)

I. 130 - 131. It is not immediately obvious why the presence of an adenine methylating type IIG system and type I system would be incompatible: this may again be due to the lack of detail in the introduction.

Both reviewers have highlighted this statement. The only other strain in the global collection of 805 strains that contains the exact same type IIG system is SRR980571 (MLST 132) and also lacks a type I RM system.

The co-presence of an adenine methylating type IIG and type I system might be incompatible due to the potential for overlap in the adenine sites methylated. (Page 7; Lines 173-174)

I. 183: I believe 'of' in 'of which' can be deleted.

“of” deleted.

I. 227: review the bold formatting (does it signify anything?) and remove the stray parenthesis.

Removed bold formatting and stray bracket.

I. 238: write 'the HsdS_2 allele'

Added “the”

I. 269: why is reference 30 cited here? Perhaps it is better to include this in the methods section?

We have modified the text and made the description concise:

A significant difference in HsdS allele proportions was observed between clade A1 and non-clade A1 strains (Chi-square test of independence, p-value <0.001 and a Chi-square statistic of 266.7; degrees of freedom =2)(30).

(Page 12; Line 340-342)

I. 295 - 299: this section is difficult to parse. The authors write that there is a correlation between the presence of HsdS and ArdA with 382 strains having at least one ArdA and HsdS allele. However a similar number (225 + 134 = 359) only have a HsdS or ArdA allele, so it is not clear whether it can be suggested that 'a correlation between the presence of these alleles' can be suggested.

Deleted: suggesting a correlation between the presence of these alleles in *E. faecium* strains.

I. 306: it is probably better to write 'similar, but not identical, to Tn916.

Modified as suggested.

ArdA2 also appears to be on a transposable element similar, but not identical to Tn916 (35).

(Page 13; Lines 378-379)

References: many references lack page numbers or article identifiers.

We have formatted the references in the ASM Journals style and it is stated in the instructions for authors that DOIs will be added by ASM copy editors on all references.

=====

Reviewer #2 (Comments for the Author):

In this work, the authors examine barriers to horizontal gene transfer in hospital-adapted *Enterococcus faecium* strains, particularly the effect of Type I restriction-modification systems and anti-restriction ArdA proteins. 805 strains were examined computationally; the transformation efficiency of 20 of these was examined experimentally; 8 of these were sequenced and their methylomes examined using the PacBio platform. Most of the 805 strains contained at least 1 Type I RM system, but methylome analysis suggested that the majority of these may be silenced. Most of the 805 strains also contained at least 1 *ardA* allele, suggesting a possible method for the silencing, but deletion experiments showed that, by and large, they are not, at least not in *E. faecium*. The mechanism of RM suppression remains unknown.

Despite not identifying the mechanism of suppression, this work adds to our knowledge of the overall pictures of transformation barriers and genome defense mechanisms in *E. faecium*. I have a few minor comments and questions.

On the one hand, one might expect fewer barriers to HGT in hospital adapted strains (A1) than in the broader environment (A2 and B), given the need for acquisition of antibiotic-resistance factors. The greater number of plasmids in A1 suggests this as well. Yet, we see a preponderance of Type I RM systems (silent or

not) in A1, even excluding the plasmid-associated HsdS_4. Do the authors have an explanation? Are hospital-adapted strains exposed to more frequent phage attacks?

We have added a sentence into the results:

Conversely, the increased prevalence of HsdS in the clinical strains could help resist phage predation or as previously suggested shape the plasmidome of specific clones by allowing plasmid transfer between strains with compatible HsdS alleles (12).

(Page 12; Lines 345-347)

There seems to be little correlation between Type I system activity and transformation efficiency, at least in the set of 8 sequenced strains. (3/8 strains show m6A methylation, 2/20 strains show poor transformation, and only one strain shows both phenotypes.) In that one case it is hsdR that is responsible [Fig 3], but methylation overall is poor, suggesting a possible phase-variable repressor. Perhaps looking for hallmarks of phase variable mechanisms (repeats, etc.) in the sequenced genome might help identify other factors at play here.

We were unable to identify any genomic features that suggest that type I RM is phase variable in the E. faecium isolates examined. It is possible that subpopulations of cells exist that express the "inactive hsdMSR" system in strain where methylation was not detected. This would be a new avenue to pursue.

In Figure 1, I find the "orange" and "light orange" very difficult to distinguish. Can they be made more different?

Thank you for identifying this issue. Figure 1 has been modified with the light orange changed to purple (ST80 isolates). All figures have been reviewed to ensure they are colour-blind accessible.

Lines 130-131. Although incompatibility can't be ruled out, it is hard to make that case based on one example.

Both reviewers have highlighted this statement. The only other strain in the global collection of 805 strains that contains a type IIG system is SRR980571 (MLST 132) and also lacks a type I RM system.

The co-presence of an adenine methylating type IIG and type I system might be incompatible due to the potential for overlap in the adenine sites methylated.

(Page 7; Lines 173-174)

Line 211. Why was strain 2397 not used as the "control" for 2394, since its HsdS_3 is stated to be 100% identical to 2394 (lines 169-70, Fig S1), and it also does not have the potentially confounding second Type I system, HsdS_4?

The strains used in AUS0233, 2394 and 2406 all contain only the HsdS_3 allele, as does strain 2397 (with 2406 also containing HsdS_4). However, due to the low transformation efficiency associated with 2397 we were unable to obtain transformants for the deletion of the HsdS_3 allele from the strain. However, AUS0233 would play the same role as a control in these experiments as it only contains the HsdS_3 allele (without methylation identified from PacBio sequencing) and the deletion of the entire type I system from AUS0233 did not impact transformation.

In Supp Fig S3, the significance of the colored circles should be noted. Are these HsdS_1-4? Also, it might be helpful to add the HsdS allele numbers as well as representative recognition sites, where known.

We have adjusted the text in the figure legend to describe the circles. The numbers of the HsdS allele (1-14) have also been added to the end of the nodes and the recognition motif where known added also. The protein alignment in Supp Fig 3B figure has been modified to improve the visualisation of similarities between alleles.

Lines 270-1. The authors seem to be arguing that the lack of hsdS alleles in non-clinical isolates is due to lower selection pressure to acquire MGEs in their environment. The converse of this would be the greater

number of hsdS alleles in the hospital environment is due to greater selection pressure to acquire MGEs, which again seems counterintuitive to me, since RM systems are generally thought of as barriers to HGT. Can this be clarified?

We have added a sentence after the description of the B/A2 clade to explain a reason for the increased presence and diversity of HsdS in clinical strains:

Conversely, the increased prevalence of HsdS in the clinical strains could help resist phage predation or as previously suggested shape the plasmidome of specific clones by allowing plasmid transfer between strains with compatible HsdS alleles.

(Page 12; Lines 337-339)

Line 298-9. Of 681 A1 strains, 86% have an ardA allele (~586). Of these, 225 lack an HsdS (38%). This does not seem like much of a correlation.

After re-reading the section, we have removed the correlation sentence and left it as a descriptive analysis of HsdS/ArdA co-occurrence.

There seems to be significant recombination between the hsdS alleles, or at least the TRDs within (Supp Fig S3). What about the hsdR and M genes? Are these largely identical between the systems?

*We have conducted further analysis of the type HsdMRS loci and have included protein alignments of HsdM, HsdS and HsdR from strains with confirmed adenine methylation from the four major HsdS alleles. These analyses show that the HsdM and HsdR alleles are different, but conserved within the HsdS alleles 1 to 4 eg. for HsdS_3 strains the M and the R are always the same proteins alleles. We have made a new supplementary figure, named **Supplementary figure 1**, with the other figure moved down one. This has been reflected in the manuscript.*

Alignment of the HsdM and HsdR proteins obtained from the four type I RM systems with confirmed methylation activity (*E. faecium* 1,231,502 – HsdS_1 (11), 2405 HsdS_2 / HsdS_4 and 2394 HsdS_3), identified that minor differences in the amino acid sequence, in comparison to substantial variation observed in HsdS (**Supplementary Figure 1**). These differences were conserved across the panel of 20 isolates containing the same HsdS alleles, except for strains 2395 and 2396 (HsdS_1) which contained an HsdR^{P1026A} change. One caveat is that due to the presence of repeating DNA sequence features contigs often break at the very 5' end of *hsdM* and/or the very 3' end of *hsdR* in Illumina only assemblies, precluding complete analysis of the region.

(Page 9; Lines 219-226)

Additionally, we identified a mistake in the colouring of two circles in Figure 2B, which has now been corrected. We have also simplified the labelling in Figure 2C, to make it less cluttered and easier to read.

We also identified the mechanism of the hsdS_1 truncation in the 3 ST80 isolates from the 20 strain panel, with the follow text added.

The HsdS_1 allele was found in three out of four ST80 isolates and two out of seven ST203 but were truncated through the insertion of a transposase in the ST80 isolates examined.

(Page 7; Line 180-182)

Re: Spectrum00289-25R1 (Comparative and functional genomic analysis of foreign DNA defence mechanisms in *Enterococcus faecium*)

Dear Dr. Ian Robertson Monk:

Your manuscript has been accepted, and I am forwarding it to the ASM production staff for publication. Your paper will first be checked to make sure all elements meet the technical requirements. ASM staff will contact you if anything needs to be revised before copyediting and production can begin. Otherwise, you will be notified when your proofs are ready to be viewed.

Sincerely,
Daria Van Tyne
Editor
Microbiology Spectrum